# A Novel Load Balancing Scheme for Satellite IoT Networks Based on Spatial–Temporal Distribution of Users and Advanced Genetic Algorithms

**DOI:** 10.3390/s22207930

**Published:** 2022-10-18

**Authors:** Wenliang Lin, Zewen Dong, Ke Wang, Dongdong Wang, Yaohua Deng, Yicheng Liao, Yang Liu, Da Wan, Bingyu Xu, Genan Wu

**Affiliations:** 1The School of Electronic Engineering, Beijing University of Posts and Telecommunications, Beijing 100876, China; 2Science and Technology on Communication Networks Laboratory, Shijiazhuang 050081, China; 3School of Information and Communication Engineering, Beijing University of Posts and Telecommunications, Beijing 100876, China; 4Academy for Network & Communications of CETC, Shijiazhuang 050081, China; 5China Academy of Information and Communications Technology, Beijing 100191, China; 6China Academy of Space Technology, Beijing 100094, China

**Keywords:** load balancing, satellite network, Internet of Things, genetic algorithm, beam hopping

## Abstract

Satellite IoT networks (S-IoT-N), which have been a hot issue regarding the next generation of communication, are quite important for the coming era of digital twins and the metaverse because of their performance in sensing and monitoring anywhere, anytime, and anyway, in more dimensions. However, this will cause communication links to face greater traffic loads. Satellite internet networks (SIN) are considered the most possible evolution road, possessing characteristics of many satellites, such as low earth orbit (LEO), the Ku/Ka frequency, and a high data rate. Existing research on load balancing schemes for satellite networks cannot solve the problems of low efficiency under conditions of extremely non-uniform distribution of users (DoU) and dynamic density variances. Therefore, this paper proposes a novel load balancing scheme of adjacent beams for S-IoT-N based on the modeling of spatial–temporal DoU and advanced GA. In our scheme, the PDF of the DoU in the direction of movement of the SSP’s trajectory was modeled first, which provided a multi-directional constraint for the non-uniform distribution of users in S-IoT-N. Fully considering the prior periodicity of satellite movement and the similarity of DoU in different areas, we proposed an adaptive inheritance iteration to optimize the crossover factor and mutation factor for GA for the first time. Based on the proposed improved GA, we obtained the optimal scheme of load balancing under the conditions of the adaptation from the local balancing scheme to global balancing, and a selection of Ser-Beams to access. Finally, the simulations show that the proposed method can improve the average throughput by 3% under specific conditions and improve processing efficiency by 30% on average.

## 1. Introduction

Sensing and monitoring anywhere, anytime, and anyway in more dimensions are quite important for the coming era of digital twins and the metaverse [1]. The internet of things (IoT) is a network that provides the methods and infrastructures needed to achieve states sensing of object and relative information aggregation. The essential tendency of IoT in the future is global coverage and remote controlling with a higher data rate and access amount. Therefore, the IoT on the ground combined with the satellite network would be the most potent approach to greatly enlarging the scenarios of the IoT [2]. With the recent rapid developments in satellite networks, satellite IoT networks (S-IoT-N) have been a topic of general interest in the monitoring of mining, ocean shipping, and power transmission.

The key bottleneck of the above urgent application is the enhancement of the satellite network. One of the scenarios of the sixth generation (6G) is the space–air–ground integrated network (SAGIN). The Third Generation Partnership Project (3GPP) have proposed non-terrestrial networks (NTN) projects to promote the enhancement of satellite networks since 2019 [3]. Satellite internet networks (SIN) are regarded as the most possible evolution road, possessing characteristics of many satellites, low earth orbit (LEO), the Ku/Ka frequency, and a high data rate [4].

More advanced potential technologies are introduced in SIN evolution. There are some important tendencies:

(a) Satellite beams (Sat-Beam) steering in a permanent position: SIN satellites have multi-Sat-Beams, which move at high speed relative to the earth. The footprint of Sat-Beams (F-Sat-Beam) are planned by constellation [5]. During the service of the observed satellite, its Sat-Beams steer in a permanent position.

(b) Intra-serving beam (Ser-Beam)-hopping (Ser-BH): For the exploitation of a satellite-based phased array antenna (PAA), the F-SB can be further divided into different small footprints of serving beams (F-Ser-Beam) [6]. They can serve users at different places and in different periods through Ser-BH, which achieves occasion division multiplexing.

(c) More dimensions of radio resource (RR): The introduction of combining the fifth generation (5G) new radio (NR) technologies, orthogonal frequency division multiplexing access (OFDM), and non-orthogonal multiple access (NOMA) would be exploited by SIN. Using the coverage of Sat-Beams, the users can be served by the multiplex with time, frequency, power, space, and occasion divisions [7].

Meanwhile, the requests of users are not distributed uniformly in the space domain, which causes the density of each F-Ser-Beam to be quite different. In the scope of Sat-Beams, particularly the Ser-Beams at the edge, it is hard for users to choose the long-suitable Ser-Beams for load balancing. As the granularity of RR increases, the above problem becomes more serious. The relatively shifting position between the position of Sub-Satellite Point (SSP) and the user also makes this worse.

Figure 1 shows the problems of load balancing for Ser-BH in SIN. As Figure 1 shows, the first problem is how to improve the local scheme for load balancing of adjacent Ser-Beams to adapt to global scenarios. For large degradation of signal fading, the load balancing between adjacent Ser-Beams around the edge of Sat-Beams should be paid more attention to. However, there are thousands of possible scenarios for adjacent Ser-Beams with different DoU. The optimal scheme is not only to solve the problem in the case of local load balancing, but also different local scenarios around the world. The second problem is which Ser-Beams the users should be accessing to under the DoU of spatiotemporal non-uniform. The SSPs of an observed satellite are not permanent and not in the center of F-Ser-Beams, which are different from other satellites and have different times. The trajectories of SSPs are also different in the Ser-Beams serving during (SD). That means the probability density of power (PDP) of the satellite signal at the same position on the ground would vary with time, and the corresponding capacity of each Ser-Beam would also change.

Existing research on load balancing schemes for satellite networks focuses on Sat-Beam scenarios without intra-Ser-BH, and users with unified uniform distribution in the time and space domains [8,9,10,11,12,13,14,15]. Optimization of satellite network routes is the prevailing design for achieving load balancing in this research. However, the DoU are spatio-temporal non-uniform; none of the research can solve the problems of global load balancing based on local balancing schemes and the selection of access to Ser-Beams under changeable DoU. Considering the DoU have a serious relationship with geography, the users of S-IoT-N can be divided into different grids on the surface of the earth. After combining the grids with global population distributions, we can obtain multi-grids with different DoU densities. We can operate the load balancing according to the density variances of users under different movement directions from one grid to another grid. Though the densities of grids are different around the world, all scenarios can be taken as a limited set for density variances of users. Then, the basic load balancing schemes become the original reference for global schemes and different times, which can be taken as the gene of the genetic algorithm (GA). After modeling and improving the mutation and crossover of the gene, it can effectively solve the problem of the adaptation from a local balancing scheme to global balancing and the selection of Ser-Beams to access.

Finally, this paper proposes a novel load balancing scheme of adjacent beams for S-IoT-N based on the modeling of spatial–temporal DoU and advanced GA [16,17,18,19]. The main contributions of this paper are summarized as follows:In contrast to existing research on load balancing under DoU of uniform, we are the first to improve these schemes by modeling the density variances of users under different moving directions. This can solve load balancing problems under spatio–temporal non-uniform DoU;Fully considering the prior periodicity of satellite movement and the similarity of DoU in different areas, we propose the adaptive inheritance iteration to optimize the crossover factor and mutation factor for GA for the first time. This can enhance the efficiency and convergence speed of GA for S-IoT-N scenarios;The Ser-BH scenario is totally new to SIN. We propose a load balancing scheme based on non-uniform spatial–temporal DoU and advanced GA, which can achieve better performance of total throughput for adjacent Ser-beams.

The rest of this paper is organized as follows: In Section 2, the relative works are introduced. In Section 3, the system model and problem formulation are described. Section 4 is devoted to the load balancing scheme of adjacent beams for S-IoT-N based on the modeling of spatial–temporal DoU and advanced GA. The simulations and analysis are discussed in Section 5. The summaries are provided in Section 6.

## 2. Related Works

Several works on satellite load balancing have been published in recent years. Their current include load balancing based on the distribution of users, load balancing based on satellite ephemeris topology, and load balancing based on Quality of Service (QoS).

The schemes of load balancing based on DoU dictate that the users decide which satellites to access by analyzing the density of equipment distributions [20,21,22,23]. W. Liu etc. proposed a routing algorithm based on segment routing for traffic return of LEO satellite networks [20]. They dynamically divide the surface of the earth into light and heavy load zones according to the relative position between gateways and the reverse slot. They then use a pre-balanced shortest path algorithm in the light load zone and use the minimum weight path defined by congestion index as the routing rule in the heavy load zone. J. Camino etc. proposed a method for optimizing the layout of satellite beams by applying mixed-integer linear programming (MILP) [21]. They designed two different sizes of spot beams to cover the area under non-uniform DoU. In their schemes, low user density areas would be covered with large-size beams, and high user density areas would be covered with small-size beams. MILP was exploited to optimize load balancing to achieve well-distributed traffic among the different beams. Syed Maaz Shahid et al. proposed a load balancing algorithm for a multi-RAT (radio access technology) network including an NTN and a TN [22]. They first offloaded the appropriate edge UEs of an overloaded cell to underutilized neighboring cells in TN. If there were any overloaded cells after the first step, they offloaded the delay-tolerant data flows of UEs to a satellite link.

The load balancing schemes based on satellite topology dictate that the users decide which satellites to access by analyzing which planned satellites would be on the services. P. Liu etc. proposed a load balancing routing scheme by hybrid-traffic-detour [24]. They calculated and designed the shortest path and a long-distance traffic detour path to forward packets and determine which path to use for forwarding to against link congestion [25]. The scheme tried to avoid a situation in which the routing path between a node and its neighbors were directed toward the same destination node. It chose the next hop based on satellite ephemeris prediction and traffic distribution, which increased the effect of load balancing. C. Dong et al. proposed a load balancing routing algorithm based on extended link states [26]. They keep all the satellite nodes informed of the link congestion state through an active state discovery and automatic congestion state release mechanism. All satellite nodes update the route table according to the link congestion state to achieve a balanced distribution of traffic load.

Load balancing schemes based on QoS dictate that the users decide which satellites to access by analyzing the average QoS of the satellite network. H. Cao et al. proposed a load balancing algorithm under the satellite network with hopping beams [27]. They divided the beams into heavy load beam (HLB) and light load beam (LLB) groups and offloaded the user terminals that had the highest packet loss rate in the HLB to an adjacent LLB. Then they offloaded the remaining user terminals that had the highest packet loss rate in the HLB to a non-adjacent LLB with hopping beams.

After reviewing the existing works on the load balancing of satellites, most research institutes were concerned with treating satellite networks as a pure transmission network. Their schemes mainly selected a suitable route to achieve efficient traffic offloading. Usually, they focused on the scenarios of the Sat-Beams without intra-Ser-BH, and the users with unified uniform distribution in the time and space domains. However, the DoU are spatio–temporal non-uniform, and none of this research has solved the problem of global load balancing based on a local balancing scheme and the selection of access to Ser-Beams under changeable DoU. On the other hand, artificial intelligence (AI) has been widely introduced to solve random communications issues.

## 3. System Model and Problem Formulations

### 3.1. Network Architecture

As is shown in Figure 2, the network architecture of S-IoT-N can be divided into space segments and ground segments. In the space segment, there are multi-satellites $S=\left\{s_{11},s_{12},s_{13},\ldots ,s_{ij},\ldots ,s_{mn}\right\}$, which are composed of a satellite constellation in orbit of height; h. $s_{ij}$ denotes a satellite, where i and j are the i-th satellite in j-th orbit in the constellation. Next, the number of satellites is expressed as m∗n. There are intra-links $l_{s_{i_{a}j_{a}}\to s_{i_{b}j_{b}}}$ between one satellite $s_{i_{a}j_{a}}$ and another satellite $s_{i_{b}j_{b}}$. The satellites provide users with communication coverage on the ground, which exploits the PAA to achieve dynamic beam-hopping. The coverage areas are further planned and configured according to the satellite constellation and the surface terrain of the earth. These consist of sets of F-Sat-Beams $B=\left\{B_{1},B_{2},B_{3},\ldots ,B_{N}\right\}$, where $B_{i}$ corresponds to the coverage area. All F-Sat-Beams cover the globe; the sum of coverages is $\sum_{0}^{N}S_{e}\left(B_{i}\right)$. In the system using Ser-BH, the coverage of F-Sat-Beam $B_{i}$ is a virtual boundary for the serving satellite on the ground. There are several Ser-Beams $b_{ij}$ in each F-Sat-Beam $B_{i}$. F-Ser-beams correspond to the actual position of the physical beams serving users. The number of F-Ser-Beams in each F-Sat-Beam can be different. The sum of the coverage area of F-Ser-Beam $\sum_{0}^{M}S_{e}\left(b_{ij}\right)$ should be larger than that of the F-Sat-Beams. For example, the OneWeb constellation is a near-polar-orbit satellite constellation comprising 720 satellites in 18 circular orbital planes at an altitude of 1200 km [15,16]. The satellite can be set as $S=\left\{s_{11},s_{12},s_{13},\ldots ,s_{ij},\ldots ,s_{\left(40\right)\left(18\right)}\right\}$. Each satellite has about 6 adjacent satellites and 16 identical, non-steerable, highly-elliptical F-Ser-Beams. The F-Ser-Beams in each F-Sat-Beam $B_{i}$ can be set as $b_{ij}\;,\;j=1\cdots 16$. If the users in the coverage area are served, the situation is assumed as $b_{ij}\left(\mathrm{on}\right)$; otherwise, the situation is $b_{ij}\left(\mathrm{off}\right)$. In the ground segment, there are several earth stations (ES) and users (U). The ESs are the nodes that receive or transmit information to or from the satellites, whose functions are similar to the functions of a base station (BS). The users are the IoT terminals, the number of them in the world is assumed as N. The users are distributed in spatio–temporal non-uniform by geography. We will further demonstrate the user model and distributions in the next chapter.

### 3.2. User Model and Distribution

The DoU are analyzed in two scopes: the global scope, and the local scope. In the global scope, the user distributions are relatively permanent and follow global population distributions. This can provide useful prior knowledge. The densities of users vary greatly between each F-Ser-Beam. In the local scope, since the coverage of F-Sat-Beams is much larger than those of cells of mobile communication networks, there are very weak mutual influences between non-adjacent Ser-Beams in S-IoT-N. Therefore, the handovers between adjacent Ser-Beams based on received signal-power are not considered in our scenarios. We are concerned with the problem of which Ser-Beams the users should access for load balancing. The solutions were further improved to adapt to different density variances around the world. Next, we proposed to model the density variances of users under different moving directions, which is shown in Figure 3.

First, we built a two-dimensional satellite to ground grid map (SGGM). In an SGGM, a plane projection $S_{u}$ of the surface of the earth is generated. $S_{u}$ can be divided into a×b grids according to the configured satellite constellation, where a is the number of grids in the latitudinal direction, and b is the number of grids in the longitudinal direction. It can be deduced as:(1)$$a=\left[a_{0}\times \frac{N_{s0}}{N_{s1}}\times \frac{H_{o1}}{H_{o0}}\right]$$
(2)$$b=\left[b_{0}\times \frac{N_{s0}}{N_{s1}}\times \frac{H_{o1}}{H_{o0}}\right]$$
where $a_{0}$ and $b_{0}$ are the number of grids in the directions of latitude and longitude in the case of the referenced satellite constellation, respectively. $N_{s0}$ and $H_{o0}$ are the satellite number and the orbital altitude of the referenced satellite constellation. $N_{s1}$ and $H_{o1}$ are the satellite number and orbit height of the observed satellite constellation. The observed grid $G_{\left(a,b\right)}$ is associated with an actual latitude and longitude, which is $\left(x_{\left(Ga_{1},Gb_{1}\right)},y_{\left(Ga_{1},Gb_{1}\right)}\right)$, $\left(x_{\left(Ga_{2},Gb_{2}\right)},y_{\left(Ga_{2},Gb_{2}\right)}\right)$, $\left(x_{\left(Ga_{1},Gb_{2}\right)},y_{\left(Ga_{1},Gb_{2}\right)}\right)$, and $\left(x_{\left(Ga_{2},Gb_{1}\right)},y_{\left(Ga_{2},Gb_{1}\right)}\right)$.

Second, the probability density functions (PDF) of DoU $p_{0}\left(x,y\right)$ of each grid were obtained by combining the SGGM and global IoT device distribution. The value of PDF is shown in the second part of Figure 3. If the number $G_{\left(a,b\right)}$ of observed satellite grids is greater than the number $G_{\left(a0,b0\right)}$ of referenced satellite grids, that is, $a_{1}>a_{0}$, $p_{1}\left(x,y\right)\;$ can be exploited directly. If the number $G_{\left(a,b\right)}$ of observed satellite grids is less than the number $G_{\left(a0,b0\right)}$ of referenced satellite grids, that is, $a_{0}>a_{1}$, the densities $p_{1}\left(x,y\right)\;$ can be gained by obtaining the average of the densities $p_{0}\left(x,y\right)$ of the surrounding grids. Next, the new PDF of DoU $p_{g}\left(G_{\left(a,b\right)}\right)$ with different specifications of the SGGM was obtained.

Third, the PDF of DoU densities $p_{g}\left(G_{\left(a,b\right)}\right)$ were further normalized. We designed 6 levels of normalized densities for $p_{g}\left(G_{\left(a,b\right)}\right)$, which are $\left\{Q_{1},Q_{2},Q_{3},Q_{4},Q_{5},Q_{6}\right\}$; $Q_{1}:p_{g}\left(G_{\left(a,b\right)}\right)\in \left[0:0.02\right]$,$Q_{2}:p_{g}\left(G_{\left(a,b\right)}\right)\in \left[0.02:0.06\right]$, $Q_{3}:p_{g}\left(G_{\left(a,b\right)}\right)\in \left[0.06:0.1\right]$, $Q_{4}:p_{g}\left(G_{\left(a,b\right)}\right)\in \left[0.1:0.2\right]$, $Q_{5}:p_{g}\left(G_{\left(a,b\right)}\right)\in \left[0.2:0.5\right]$, and $Q_{6}:p_{g}\left(G_{\left(a,b\right)}\right)\in \left[0.5:1\right]$.

Finally, the PDF variances with moving SSPs were obtained. Under the observation of SSP movement directions, the differences $∆p_{g}\left(x,y\right)$ in PDF during a Sat-Beam serving period $T_{d}$ from one grid to another grid are expressed as
(3)$$∆p_{g}\left(G_{\left(a,b\right)}\right)=p_{g}\left(G_{\left(a_{i},b_{i}\right)}\right)-p_{g}\left(G_{\left(a_{i+v_{dT_{d}}},b_{i}+i*v_{dT_{d}}\right)}\right).$$

### 3.3. Problems Formulations

In Equation (3), the PDF variances of different grids changing around the SGGM can be described by $∆p_{g}\left(x,y\right)$ to provide a foundation for solving the problems of global load balancing adaption. Here, the problems should be further analyzed in depth. In actual S-IoT-N scenarios, users are not distributed uniformly in different grids, especially for non-centered SSPs. As shown in Figure 4, two adjacent F-Sat-Beams $B_{1}$ and $B_{2}$ were observed. $B_{1}$ can be further divided into several F-Ser-Beams $\left\{b_{11},b_{12},\ldots ,\;b_{1n}\right\}$; $B_{2}$ can be further divided into several F-Ser-Beams $\left\{b_{21},b_{22},\ldots ,\;b_{2n}\right\}$. The Ser-Beams $b_{edge}\left(B_{1},B_{2}\right)\;$ around the edge of $B_{1}$ and $B_{2}$ are of most concern. The serving periods $T_{n}$ of different satellites in the satellite constellation serving on the same F-sat-Beam are different, which have different start-serving-times $t_{s}$ and over-serving-times $t_{o}$. Figure 4 shows the trajectories of the SSPs in F-Ser-Beams $B_{1}$ and $B_{2}$ at the periods of $T_{1}$ and $T_{2}$, which are set as $l_{11}@B1\&T1$ and $l_{21}@B2\&T1$. Next, in the period $T_{1}$, the center-point of $b_{1e}\in b_{edge}\left(B_{1},B_{2}\right)$ is closer to the SSP trajectory $l_{21}$ of $B_{2}$ than SSP trajectory $l_{11}$ of $B_{1}$. In the period $T_{2}$, the center-point of $b_{1e}$ is closer to the SSP trajectory $l_{12}$ of $B_{1}$ than the SSP trajectory $l_{22}$ of $B_{2}$. Considering that the distances between the center-points of Ser-Beams and satellites are similar, the SSPs in the two Ser-Beams are similar too. Therefore, the problem of load balancing lies in determining which satellite’s Ser-Beams users should access, and not the access threshold. Following this problem, it can be analyzed that the constraints of optimal load balancing relate to the density variances of DoU with the SSP movement, the number of users with different services, and the differences in DoU of adjacent F-Ser-Beams.

During a specific serving period $T_{i}$, Sat-Beam $B_{i}$ was observed in the analysis. The maximum capacity of a satellite was assumed as $Tra_{max}\left(B_{i}\right)$, the total bandwidths as $BW_{i}$, and the DoU density as $p_{g}\left(B_{i}\;,T_{i}\;,t\right),t\in \left[T_{i-1}\;,T_{i}\right]$. The Ser-Beams of $B_{i}$ are $b_{B_{i}}=\left\{b_{i1},b_{i2},\ldots ,\;b_{in}\right\}$, the area of each Ser-Beam is s, and the corresponding PDF of DoU is $p_{g}\left(b_{ik}\right),k=1\cdots \;n$. The average arrival probability of services requests is $\mu_{i}\left(t\right)\;,t\in \left[T_{i-1}\;,T_{i}\right]$. The average service traffic is $\varphi_{i}$. The service traffic of Ser-Beams is,
(4)$$Tra\left(b_{ik}\right)=\iint_{S}p_{g}\left(b_{ik}\right)\times \mu_{i}\left(t\right)\times \varphi_{i}\;.$$

The adjacent Sat-Beam of $B_{i}$ is $B_{j}$. The adjacent Ser-Beams are $b_{m}\in b_{edge}\left(B_{i},B_{j}\right)$. The judgment coefficient function $\lambda \left(b_{m},*\right)$ is defined as
(5)$$\lambda \left(b_{m},B_{i}\right)=\left\{\begin{array}{c} 1,\;users\;in\;beam\;b_{m}\;access\;to\;satellite\;B_{i}\\ 0,\;users\;in\;beam\;b_{m}\;not\;access\;to\;satellite\;B_{i} \end{array}\right.$$

Next, the traffic of the Sat-Beam is,
(6)$$Tra\left(B_{i}\right)=\underset{b_{ik}\in b_{B_{i}}}{\sum }Tra\left(b_{ik}\right)+\underset{b_{m}\in b_{edge}\left(B_{i},B_{j}\right)}{\sum }\lambda \left(b_{m},\;B_{i}\right)\times Tra\left(b_{m}\right)-\underset{b_{ik}\in b_{B_{i}}\cap b_{edge}\left(B_{i},B_{j}\right)}{\sum }Tra\left(b_{ik}\right)$$

We can also obtain the traffic of Sat-Beam $B_{j}$ with $Tra\left(B_{j}\right)$.

Finally, the problem of load balancing for adjacent Ser-Beams of S-IoT-N can be formulated as:Objectives:

Objective 1: achieve the maximum throughput of adjacent Sat-Beams:(7)$$\mathrm{MAX}\;\left[Tra\left(B_{i}\right)+\sum Tra\left(B_{j}\right)\right]\;$$

Objective 2: achieve the minimum waiting time for users of adjacent Sat-Beams:(8)$$\mathrm{MIN}\left[average\left(Tra\left(b_{ik}\right)/BW_{i}\right)\right]$$

Conditions:

Condition 1: The PDF variances of DoU in the direction of SSPs moving in a Sat-Beam $B_{i}$ and serving time $T_{i}$ are observed.
(9)$$∆p_{g}\left(B_{i}\;,T_{i}\;,t\right)=\frac{\partial p_{g}\left(B_{i},T_{i},t\right)}{\partial t}\;,t\in \left[T_{i-1}\;,T_{i}\right]$$

Conditions 2: The PDF variances of DoU in the direction of SSPs moving between two adjacent Sat-Beams in one serving time $T_{i}$ are observed. The DoU of two adjacent F-Sat-Beams is different.
(10)$$∆p_{g}\left(B_{i}\;,B_{j}\;,T_{i}\;,t\right)=\frac{\partial p_{g}\left(B_{i}\;,T_{i}\;,t\right)}{\partial t}-\frac{\partial p_{g}\left(B_{j}\;,T_{i}\;,t\right)}{\partial t}\;,t\in \left[T_{i-1}\;,T_{i}\right]$$

The probability of judgment coefficient function can be varied with density variances, which are,
(11)$$\ln\frac{P\left(\lambda \left(b_{m}\lambda B_{i}\right)=1\right)}{P\left(\lambda \left(b_{m}\lambda B_{j}\right)=1\right)}=\frac{\varepsilon }{∆p_{g}\left(B_{i}\;,B_{j}\;,T_{i}\;,t\right)}$$
where $∆p_{g}\left(B_{i}\;,B_{j}\;,T_{i}\;,t\right)>0$, indicating that the density variances of $B_{i}$ increase more than those of $B_{j}$. Thus, the probability of accessing $B_{i}$ decreases, achieving load balancing.

Condition 3: The density variances of DoU in the direction of SSPs moving between two adjacent Sat-Beams in two serving times $T_{i}$ are observed. The trajectories of SSPs are different.

Constraints:

Constraint 1: The total traffic of users newly accessing the adjacent Sat-Beams is less than the traffic capacity of the Sat-Beams:(12)$$Tra\left(B_{i}\right)\leq Tra_{\max}\left(B_{i}\right)$$
(13)$$Tra\left(B_{j}\right)\leq Tra_{\max}\left(B_{j}\right)\;$$

## 4. Load Balancing Scheme Based on the Modeling of Spatial–Temporal DoU and Advanced Genetic Algorithms

### 4.1. The Modeling of the Solution Using the Original GA

In Section 3, the constraints of the problems of load balancing were proposed. A local optimal solution must be achieved to solve the problems of most other cases of load balancing around the world. At the same time, the local optimal solution should be further improved to adapt to the density variances of DoU in the spatial–temporal domain. In addition, the periodic movement of satellites can provide solution optimization with more efficient prior information. Based on the above considerations, the optimal solution was modeled by GA. The overall design of the original GA is shown in Table 1.

The specific modeling is described as follows:Step 1: Modelling of Fitness Function:

The fundamental purpose of load balancing for S-IoT-N is to improve the performance of the throughput and QoS. In serving period $T_{i}$, the throughput of all Sat-Beams $B_{j}$ should be maximized, and the average waiting time of all users should be minimized. The fitness function is defined as
(14)$$fitness=\frac{k_{1}*Tra_{total}}{k_{2}*T\_delay_{ave}}$$
where $Tra_{total}$ and $T\_delay_{ave}$ are the global throughput of S-IoT-N and the average waiting time of all users.

Step 2: Modelling of the Solution:

As is shown in Figure 5, we assume that in the observed area, there are k Sat-Beams $B_{ij}$, and the corresponding PDF of DoU is $p\left(B_{ij}\right)$. A solution $A_{ij}$ to the problem of load balancing, which achieves the optimization objective, is set as a single optimal solution output by GA. $A_{ij}$ are encoded in the form of natural numbers, which is a matrix composed of *k* natural numbers, and $a_{m}$ denotes the m-th access scheme of the m-th Ser-Beam. Under the possible values of each solution $a_{m}$, the influence factors are the density variances of the user $∆p_{g}\left(B_{ij},T_{i}\right)$ of the serving satellite and the difference in the density of adjacent satellite users $∆p_{g}\left(B_{ij}\;,B_{i+∆i\;j+∆j}\;,T_{i}\right)$ within the period $T_{i}$ instead of traditional random influence for the schemes to which Sat-Beams have access to. This can achieve a certain optimization direction in a specific scenario.

Under the above optimization direction, we have the solutions $A_{ij}^{1}\;,A_{ij}^{2}\;,A_{ij}^{3}\;,\cdots ,A_{ij}^{n}$, which are the initial populations of GA optimization. The offspring individuals generated by the crossover mutation in each generation and the parent individuals form a new group and N individuals $A_{ij}^{1}\;,A_{ij}^{2}\;,A_{ij}^{3}\;,\cdots \;,A_{ij}^{N}$ are selected using the tournament method to form the next generation populations. The new populations participate in the evolution of the next generation.

Step 3: Modelling the Genetic Cross:

According to the individual pairing principle of matching each other, two individuals with similar fitness, $A_{ij}^{x}$ and $A_{ij}^{y}$, are selected. We judge whether the crossover occurs according to the crossover probability $P_{acr}$. A floating-point number is randomly generated in the interval [0,1] as the judgment factor α. If $\alpha \leq P_{acr}$, performs the crossover operation, randomly select l consecutive elements (l<k) in the individual $A_{ij}^{x}$, and exchange them with the elements in the same position in $A_{ij}^{y}$ to generate two offspring individuals. The probability of genetic crossover is defined as follows:(15)$$P_{acr}=\left\{\begin{array}{c} P_{acr}\left(1\right),fitness\leq fitness_{ave}\\ P_{acr}\left(0\right)\times \frac{fitness_{max}-fitness}{fitness_{max}-fitness_{ave}}\;,else \end{array}\right.$$
where fitness is the fitness of individuals with higher fitness in an individual $A_{ij}^{x}$ and $A_{ij}^{y}$, $fitness_{ave}$ is the average fitness of the contemporary population, $fitness_{max}$ is the fitness of the best individual in the contemporary population, and $P_{acr}\left(1\right)$ and $P_{acr}\left(0\right)$ are probability constants.

Step4: Modelling Genetic Mutation:

For every individual, judge whether there is a genetic mutation according to the mutation probability $P_{mut}$. A floating-point number is randomly generated in the interval [0,1] as the judgment factor β. If $\beta \leq P_{mut}$, the mutation operation is performed, and the mutation point is randomly selected in the individual $A_{ij}^{x}$. Next, the elements of the corresponding point of $A_{ij}^{x}$ are changed according to the selection probability in the value set to generate a child individual. The probability of genetic mutation is defined as follows:(16)$$P_{mut}=P_{mut}\left(0\right)\times \frac{t_{max}-t}{t_{max}},$$
where $t_{max}$ is the maximum evolution generation of the genetic algorithm, t is the current evolution generation, and $P_{mut}\left(0\right)$ is a probability constant.

### 4.2. GA Optimization by Improving the Genetic Crossover and Mutation Based on Controllable Adaptation to the Scenario

Since the load balancing of S-IoT-N is continuous, we should fully consider the prior periodicity of satellite movements and the similarity of DoU in a different area. This can provide efficient prior information and adjust the GA according to the differences in load balancing performance between the current period and the previous period. This can control the GA’s adaptation to changes in different scenario parameters to achieve better evolution to obtain better performance. Therefore, this paper proposes a controllable adaptive genetic algorithm (CAGA) based on prior information on S-IoT-N. The optimization is mainly on the improvement of genetic crossover and genetic mutation.

Optimization of Cross Factor:

Before the service period $T_{i}$ of satellite $S_{i}$ starts, the incoming serving satellite obtained the optimal solution $A_{ij}\left(T_{i-1}\right),A_{ij}\left(T_{i-2}\right),A_{ij}\left(T_{i-3}\right),\cdots \;,A_{ij}\left(T_{i-h}\right)$ of the load balancing scheme for Ser-Beams in former h periods. They grouped a set $A\_pro_{ij}$ with a prior high-fitness cross object sequentially, and the PDF variances $∆p_{g}\left(B_{i}\;,T_{i-\delta }\;,t\right),\delta =1\cdots \;h$ of each optimal solution in the corresponding period were also obtained.

At the beginning of the crossover operation, individuals in the contemporary population were divided into individuals whose fitness is higher than the average fitness of the population and individuals whose fitness is lower than the average fitness of the population. Here, whether to cross was judged according to the crossover probability shown in Equation (15). Individuals with higher fitness than the population average were selected as crossover objects with similar fitness according to the principle of matching each other. The individuals whose fitness was lower than the average fitness of the population selected cross objects from the prior high-fitness cross object set $A\_pro_{ij}$, where the probability of each element of $A\_pro_{ij}$ being selected is determined by the differences factor of densities variances $\omega \left(T_{i}\;,T_{i-\delta }\right),\delta =1\cdots \;h$. The difference factor of PDF variances can be calculated as,
(17)$$\begin{array}{l} \;\frac{1}{\omega \left(T_{i}\;,T_{i-\delta }\right)}=\int_{T_{i-1}}^{T_{i}}{\left[∆p_{g}\left(B_{i}\;,T_{i}\;,t\right)-∆p_{g}\left(B_{i}\;,T_{i-\delta }\;,t\right)\right]}^{2}dt,\\ =\int_{T_{i-1}}^{T_{i}}{\left[\frac{\partial p_{g}\left(B_{i}\;,T_{i}\;,t\right)}{\partial t}-\frac{\partial p_{g}\left(B_{i}\;,T_{i-\delta }\;,t\right)}{\partial t}\right]}^{2}dt\\ =\int_{T_{i-1}}^{T_{i}}{\left[\frac{\partial \left(p_{g}\left(B_{i}\;,T_{i}\;,t\right)-p_{g}\left(B_{i}\;,T_{i-\delta }\;,t\right)\right)}{\partial t}\right]}^{2}dt\\ =\int_{T_{i-1}}^{T_{i}}{\left[\frac{\partial ∆p_{g}\left(B_{i}\;,T_{i}\;,T_{i-\delta }\;,t\right)}{\partial t}\right]}^{2}dt \end{array}$$

The probability of $A_{ij}\left(T_{i-\delta }\right)$ selected can be calculated:(18)$$P\left(A_{ij}\left(T_{i-\delta }\right)\right)=\frac{\omega \left(T_{i}\;,T_{i-\delta }\right)}{\sum_{d=1}^{h}\omega \left(T_{i}\;,T_{i-d}\right)}$$
where $\frac{1}{\omega \left(T_{i}\;,T_{i-\delta }\right)}$ is the quantification of the differences in user density mutation of Ser-Beams in two periods. When $\frac{1}{\omega \left(T_{i}\;,T_{i-\delta }\right)}$ became smaller, the differences of two scenarios in two periods became fewer, and the probability that the load balancing scheme of this period was selected increased.

Optimization of Mutation Factor:

Before the service period $T_{i}$ of satellite $S_{i}$ starts, the incoming serving satellite obtained $p_{g}\left(B_{i}\;,T_{i}\;,t\right)$ of F-Sat-Beams and $p_{g}\left(B_{j}\;,T_{i}\;,t\right)$ of the adjacent F-Sat-Beam which could be accessed. In the set of any feasible solution $A_{ij}=\left\{a_{1}\;,a_{2}\;,a_{3}\;,\cdots a_{k}\right\}$, the solution $a_{m}$ to access m-th F-Sat-beams included all possible F-Ser-beams in the m-th Sat-Beam. In the operation of mutation, the mutated value of $a_{m}$ was no longer randomly selected from a random probability set instead of a certain probability. The factor $\omega \left(B_{i}\right)$ that affects probability can be calculated as
(19)$$\frac{1}{\omega \left(B_{i}\;\right)}=\frac{1}{T_{i}-T_{i-1}}\times \int_{T_{i-1}}^{T_{i}}p_{g}\left(B_{i}\;,T_{i}\;,t\right)dt$$

Furthermore, the probability of the selection of scheme access to $B_{i}$ is:(20)$$\begin{array}{ll} P\left(B_{i}\right) & =\frac{\omega \left(B_{i}\right)}{\omega \left(B_{i}\right)+\omega \left(B_{j}\right)}\\ & =\frac{\frac{1}{T_{i}-T_{i-1}}\times \int_{T_{i-1}}^{T_{i}}p_{g}\left(B_{i}\;,T_{i}\;,t\right)dt}{\frac{1}{T_{i}-T_{i-1}}[\int_{T_{i-1}}^{T_{i}}p_{g}\left(B_{i}\;,T_{i}\;,t\right)dt+\int_{T_{i-1}}^{T_{i}}p_{g}\left(B_{j}\;,T_{i}\;,t\right)dt]}\;\\ & =\frac{\frac{1}{T_{i}-T_{i-1}}\times \int_{T_{i-1}}^{T_{i}}p_{g}\left(B_{i}\;,T_{i}\;,t\right)dt}{\frac{1}{T_{i}-T_{i-1}}[\int_{T_{i-1}}^{T_{i}}p_{g}\left(B_{i}\;,T_{i}\;,t\right)dt+\int_{T_{i-1}}^{T_{i}}p_{g}\left(B_{j}\;,T_{i}\;,t\right)dt]}\\ & =\frac{\int_{T_{i-1}}^{T_{i}}p_{g}\left(B_{i}\;,T_{i}\;,t\right)dt}{\int_{T_{i-1}}^{T_{i}}\left[p_{g}\left(B_{i}\;,T_{i}\;,t\right)+p_{g}\left(B_{j}\;,T_{i}\;,t\right)\right]dt} \end{array}$$

In Equation (20) we can conclude that the higher the PDF of DoU of F-Sat-Beams, the lower the probability of being selected. The overall evolution direction of the mutation operation is limited to optimization in the load balancing direction. In addition, with the observed periods and F-Ser-Beams varying, the evolution direction of the mutation can automatically adapt to real scenarios following the variances of the relative differences of PDF of adjacent F-Sat-Beams.

### 4.3. Advanced Load Balancing Scheme Based on Optimized GA

In the first two parts of this Section, we modeled the solution using the original GA and improved genetic crossover and mutation. According to this work, we designed an advanced load balancing scheme based on an optimized GA (LB-CAGA) in this part. Before the start of each satellite service period, the service satellite inputs the relevant prior information and user information of Ser-Beams and sets parameters for the GA. Next, the GA figures out the traffic load balancing scheme for the service period. The specific solution process is as follows:

The pseudocode of the algorithm is shown in Algorithm 1:

**Algorithm****1**: Load balance based on GA
1.
**Input:**

$Tra\left(b_{ij}\right);BW_{i};P\left(B_{i}\right);A_{ij}\left(T_{i-\delta }\right),P\left(A_{ij}\left(T_{i-\delta }\right)\right),\delta =1\cdots \;h;$

2.
**Output:**

$A_{ij}\_best,fitness\_best$

3.Create a population matrix $Population\left[N,k\right]$;
4.**for**i**from** 1 **to**
*N*
5.$Population\left[i,:\right]$$\leftarrow \mathrm{rand}\;(A_{ij}$)
6.

$Tra_{total}\leftarrow \sum_{j=1}^{k}Tra\left(b_{ij}\right)\times Population\left[i,k\right]$

7.

$T\_delay_{ave}\leftarrow \frac{Tra_{total}}{BW_{i}\times k}$

8.

$fitness\left(i\right)\leftarrow \frac{k_{1}\times Tra_{total}}{k_{2}*T\_delay_{ave}}$

9.
**end for**
10.Row vectors in matrix $Population\left[N,k\right]$ sorted by fitness in descending order11.

$fitness_{ave}=\frac{1}{N}\times \sum_{i=1}^{N}\;fitness\left(i\right)$

12.Perform crossover operations
$Crossover\;\left(Population\left[i,:\right]\right)$
13.Perform mutation operations $Mutation\left(Population\left[i,:\right]\right)$
14.**for**i**from** 1 **to** end15.

$Tra_{total}\leftarrow \sum_{j=1}^{k}Tra\left(b_{ij}\right)\times Population\left[i,k\right]$

16.

$T\_delay_{ave}\leftarrow \frac{Tra_{total}}{BW_{i}\times \sum_{j=1}^{k}Population\left[i,k\right]}$

17.

$fitness\leftarrow \frac{k_{1}\times Tra_{total}}{k_{2}*T\_delay_{ave}}$

18.
**end for**
19.Row vectors in matrix $Population\left[end,k\right]$ sorted by fitness in descending order 20.Tournament selects N rows to form a new population matrix21.**if** reach the maximum evolution generation **then**
22.

$A_{ij}\_best\leftarrow Population\left[1,:\right]$

23.

$fitness\_best\leftarrow fitness\left(Population\left[1,:\right]\right)$

24.
**else**
25.Go to step 1126.
**end if**



The flow chart of the algorithm is shown as Figure 6:

The load balancing scheme inputs the service traffic of the Ser-Beams, satellite total bandwidths, the optimal solution of the load balancing scheme in the former h periods, the selection probability of optimal solution, and the selection probability of a single gene. The GA is initialized first, creates a population matrix $Population\left[N,k\right]$, and generates an initial population. Next, it calculates individual fitness and the average fitness of the group. Evolution starts when initialization is complete: First, it sorts the individuals in descending order of fitness. Second, crossover and mutation operations are performed. The generated new individuals join the original population to form a new group. Third, the fitness of the new individuals and the average fitness of the new group are calculated. Finally, the population of the next generation is selected by tournament. If this evolution generation reaches the maximum evolution generation, it outputs the optimal solution and its fitness. Otherwise, it enters the next generation evolution and repeats the crossover mutation operation.

The crossover module inputs population matrix $Population\left[N,k\right]$, the fitness of all the individuals, the average fitness of the population $fitness_{ave}$, the optimal solution $A_{ij}\left(T_{i-\delta }\right)$ of load balancing scheme in the former h periods, and the selection probability $P\left(A_{ij}\left(T_{i-\delta }\right)\right)$ of the optimal solution. The crossover module judges the crossover probability and crossover mode of the individuals according to individual fitness. The individual whose fitness is greater than the average fitness of the population takes the individual behind itself as the crossover object and randomly selects the crossover starting point and crossover length. After crossover, the two sub-individuals generated are inserted at the end of the population. Individuals whose fitness is less than the average fitness of the group select the optimal solution of the load balancing scheme in the former h periods as the crossover object according to the probability and randomly select the crossover starting point and crossover length. After crossover, the two sub-individuals generated are inserted at the end of the population. Finally, the population that has completed all crossover operations is outputted.

The specific process of the crossover operation is shown in Algorithm 2:

**Algorithm 2**: Crossover
1.**Input:** $Population\left[N,k\right]$$;fitness_{ave}$$;fitness\left(i\right)$$;\;A_{ij}\left(T_{i-\delta }\right),P\left(A_{ij}\left(T_{i-\delta }\right)\right),\delta =1\cdots \;h;$2.**Output:**$Population\left[end,k\right]$;3.

i=1

4.
**while**

i≤

*N*
**do**
5.**if**$fitness\left(i\right)\geq fitness_{ave}$ **then**6.**if**$\mathrm{rand}\leq P_{acr}\left(0\right)\times \frac{fitness_{max}-fitness}{fitness_{max}-fitness_{ave}}$ **then**7.Randomly choose a crossover starting node8.Randomly choose the crossover length9.$Population\left[i,k\right]$ perform a crossover operation with
$Population\left[i+1,k\right]$10.Sub-individuals are inserted at the end of the population11.

i=i+2

12.
**else**
13.

i=i+1

14.
**end if**
15.
**else**
16.**if**$\mathrm{rand}\leq P_{acr}\left(1\right)$ **then**17.Select prior high fitness crossover objects by selection probability18.Randomly choose a crossover starting node19.Randomly choose the crossover length20.$Population\left[i,k\right]$ perform a crossover operation with $A_{ij}\left(T_{i-\delta }\right)$21.Sub-individuals are inserted at the end of the population22.

i=i+1

23.
**else**
24.

i=i+1

25.
**end if**
26.
**end if**
27.
**end while**



The mutation module inputs population matrix $Population\left[N,k\right]$, the current evolution generation t, the maximum evolution generation $t_{max}$, and the probability of single gene selection $P\left(B_{i}\right)$. The mutation module judges whether the individual mutates according to the adaptive mutation probability. The mutated individual randomly selects the mutated gene node and selects the mutated gene value according to single-gene selection probability. After mutation, the sub-individual generated is inserted at the end of the population. The population that has completed all mutation operations is outputted.

The specific process of the mutation module is shown in Algorithm 3:

**Algorithm 3**: Mutation
1.**Input:**$Population\left[N,k\right]$; t; $t_{max}$; $P\left(B_{i}\right)$;2.**Output:**$Population\left[end,k\right]$;3.

i=1

4.
**while**

i≤

*N*
**do**
5.
**if**

$\mathrm{rand}\leq P_{mut}\left(0\right)\times \frac{t_{max}-t}{t_{max}}$

**then**
6.Randomly select gene node for mutation7.Select the result of mutation by probability8.The sub-individual is inserted at the end of the population9.
**end if**
10.

i=i+1

11.
**end while**



## 5. Simulation and Analysis

### 5.1. Simulation and Experimental Design

To verify the performance of the proposed scheme, we constructed a platform to simulate the load balancing of S-IoT-N. This platform consisted of four modules: the satellite simulation module, user simulation module, coverage simulation module, and control and analysis module, as per Figure 7. The satellite simulation module was mainly used to simulate satellite nodes under different satellite configurations, which affected the size of satellite signal coverage on the ground, as well as the SSPs. The user simulation module was mainly used to simulate different DoU, which related to the population distribution and service behavior. The coverage simulation module mainly simulates F-Sat-Beams, F-Ser-Beams, the trajectory of SSPs on the ground, and the constraint boundary corresponding to the population distribution and determined constellation. The control and analysis module was mainly used to complete scenario realization, parameter configuration, and algorithm selection, as well as analysis of the learning ability of the algorithm, the throughput, and the efficiency of the traffic balancing method.

Using the simulation platform, we verified the performance of the GA’s learning efficiency and load balancing.

The parameters of GAs are configured in Table 2.

The parameters of load balancing for S-IoT-N are configured in Table 3. Load balancing simulation scenario considering 9 F-Ser-Beams at the edge of 3 F-Sat-Beams. Similar parameters can be found in the article [20].

### 5.2. Performance of the Improved Genetic Algorithm and Analysis

The solution efficiency and optimization effect of the three genetic algorithms were simulated and compared, and the statistics of the simulation results are as follows:

It can be seen from the statistical chart that under the condition of different population sizes in Figure 8, CAGA has higher optimization solution efficiency. The reason for this is that CAGA exploited prior information to limit the convergence direction of the algorithm, which accelerated the convergence speed of GA. The PAGA and traditional GA only choose better solutions in the selection module. However, the CAGA chose better solutions in the selection module, crossover module, and mutation module, which make the CAGA more efficient. Taking a population size of 60 as an example, the CAGA figured out the global optimal solution in the 1000th generation, which is 400 generations less than the probability adaptive genetic algorithm (PAGA), and nearly 1000 generations less than the traditional GA. Under the conditions of four different population sizes, the average generation used by the CAGA to search for the optimal solution is 400 generations less than that of the PAGA on average, and 700 generations less than that of the traditional GA.

### 5.3. The Performance of Load Balancing for S-IoT-N and Analysis

In the above-mentioned simulation environment of satellite traffic load balancing, LB-CAGA is compared with the most widely studied integrated weighted access scheme (IWAS). The comparison of load balancing effects of the two schemes is shown in Figure 9.

As we can see from Figure 9, the system throughput of LB-CAGA and IWAS is consistent when the user density of the adjacent F-Sat-Beam is 0.5 and 0.6. When the user density of the adjacent F-Sat-Beam is 0.7, and the user density of the target F-Sat-Beam is less than 0.74, the system throughput of LB-CAGA and IWAS is consistent. When the user density of the adjacent F-Sat-Beam is 0.7, and the user density of the target F-Sat-Beam is more than 0.74, the system throughput of LB-CAGA is on average 0.4G higher than that of IWAS. When the user density of the adjacent F-Sat-Beam is 0.8, and the user density of the target F-Sat-Beam is less than 0.62, The system throughput of LB-CAGA and IWAS is consistent. When the user density of the adjacent F-Sat-Beam is 0.8, and the user density of the target F-Sat-Beam is greater than 0.62 and less than 0.82, the system throughput of LB-CAGA is on average 0.4G higher than that of IWAS. When the user density of the adjacent F-Sat-Beam is 0.8, and the user density of the target F-Sat-Beam is more than 0.82, the system throughput of LB-CAGA and IWAS is consistent.

From the above analysis, we found that LB-CAGA does not achieve more system throughput than IWAS when the user density of F-Sat-Beam is too low or too high but achieves more system throughput when the user density of the F-Sat-Beam is high and still within the capacity of the serving satellite. The reasons are as follows:

When the user density of the F-Sat-Beam is low, the traffic load is still within the capacity of the serving satellite. Both schemes allow for the serving satellite to fully accommodate all user traffic. Consequently, the system throughput of both schemes is the same as the total traffic load. When the user density of the F-Sat-Beam is too high, the traffic load exceeds the capacity of the service satellite. Neither of the two schemes can make the serving satellite bear more traffic load. As a result, the system throughput of both schemes is the same as the capacity of the serving satellite. When the user density of F-Sat-Beam is high but still within the capacity of serving satellite, LB-CAGA can improve the total throughput of the system to a certain extent and maximize the efficiency of the satellite system. Under this condition, LB-CAGA has a better load-balancing effect than IWAS.

## 6. Conclusions

This paper proposes a novel load balancing scheme of adjacent beams for S-IoT-N based on the modeling of spatial–temporal distribution of users and advanced GAs. The main conclusions are as follows:

(1) We modeled the PDF of DoU in the direction of movement of SSP trajectory, which provided a multi-directional constraint for non-uniform distribution users in S-IoT-N. Compared with the existing model with permanent global distribution and pure random distributions, the proposed model can characterize the PDF variances of DoU more correctly in the scenario of a highly dynamic satellite.

(2) The crossover factor and mutation factor in GAs are proposed to optimize with prior information such as the periodicity of satellite movement and the proposed model of DoU, which can better improve the efficiency of GAs and the performance of load balancing in S-IoT-N than other existing methods.

(3) Based on the proposed improved GA, we obtained the optimal load balancing scheme under the conditions of adaptation from the local balancing scheme to global balancing and the selection of Ser-Beam access. In the scenario of extremely non-uniform DoU and dynamic density variances, advances in beam-hopping were fully realized.

The proposed load balancing scheme has many application scenarios in S-IoT-N. For example, S-IoT-N can make up for the lack of terrestrial networks caused by earthquakes and other natural disasters; this scheme can effectively relieve the sudden increase of traffic load pressure and keep the communication link unblocked. Moreover, when using sensors to monitor and transmit the circuit status information through S-IoT-N, this scheme can ensure the timeliness of the information, so as to take timely measures for various emergencies.

## Figures and Tables

**Figure 1 sensors-22-07930-f001:**
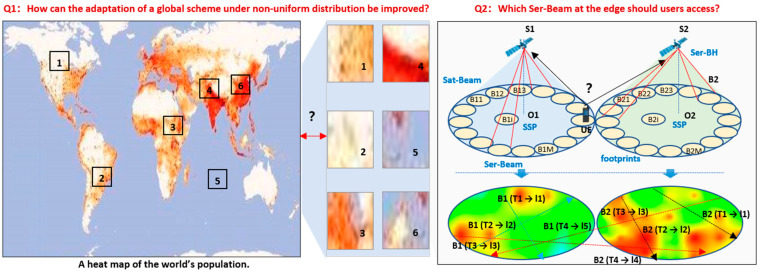
The problems of load balancing for Ser-BH in SIN.

**Figure 2 sensors-22-07930-f002:**
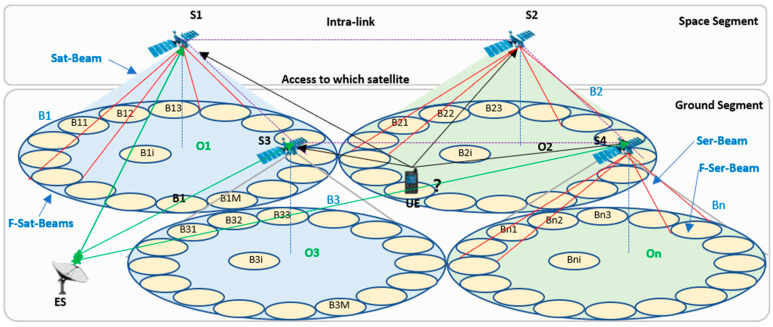
The network framework of S-IoT-N.

**Figure 3 sensors-22-07930-f003:**
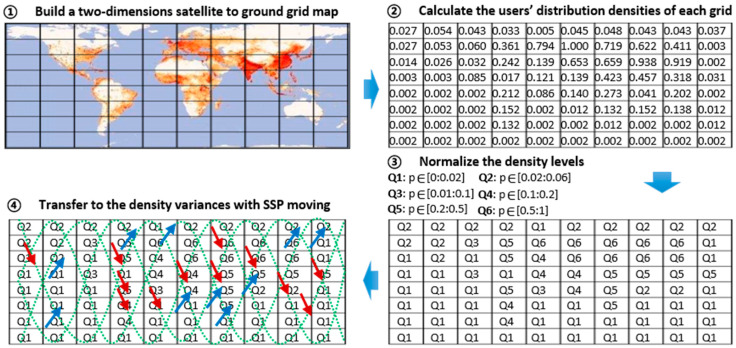
The user model and distribution of S-IoT-N.

**Figure 4 sensors-22-07930-f004:**
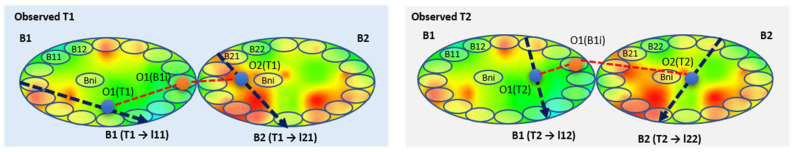
The case of different results at different times and F-Ser-Beams.

**Figure 5 sensors-22-07930-f005:**
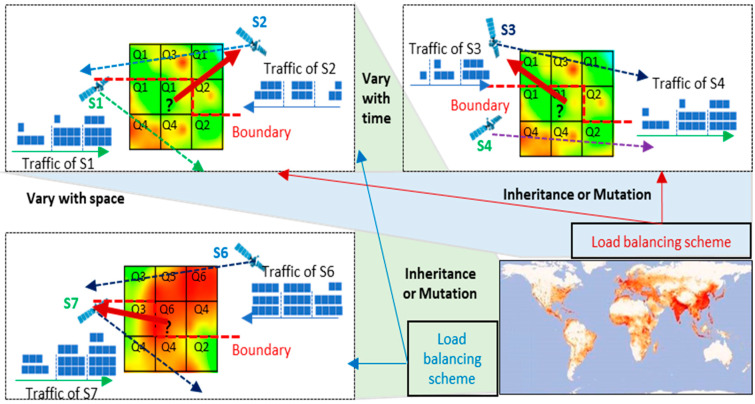
Inheritance or mutation of genetic control considering temporal variation.

**Figure 6 sensors-22-07930-f006:**
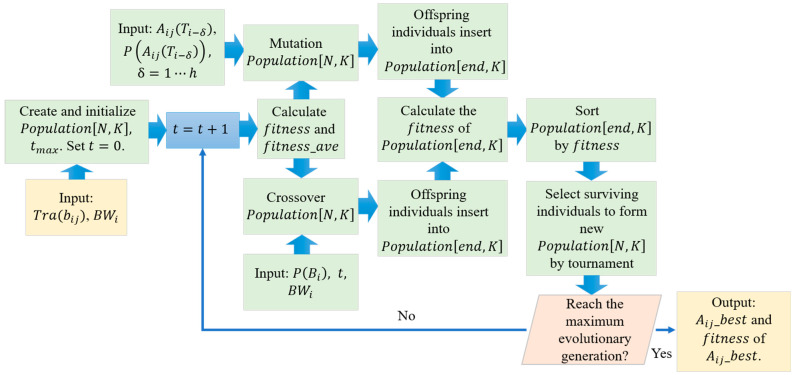
Flow Chart of load balancing method for satellite IoT beam.

**Figure 7 sensors-22-07930-f007:**
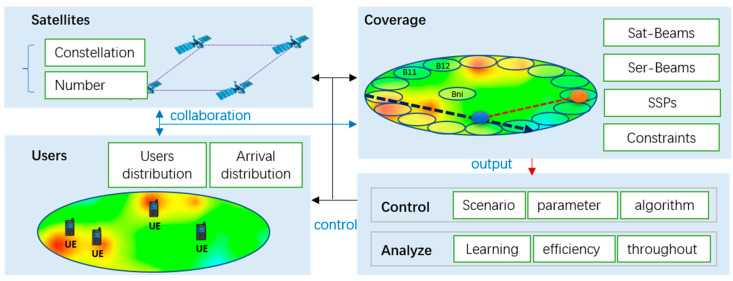
The simulation platform of load balancing for S-IoT-N.

**Figure 8 sensors-22-07930-f008:**
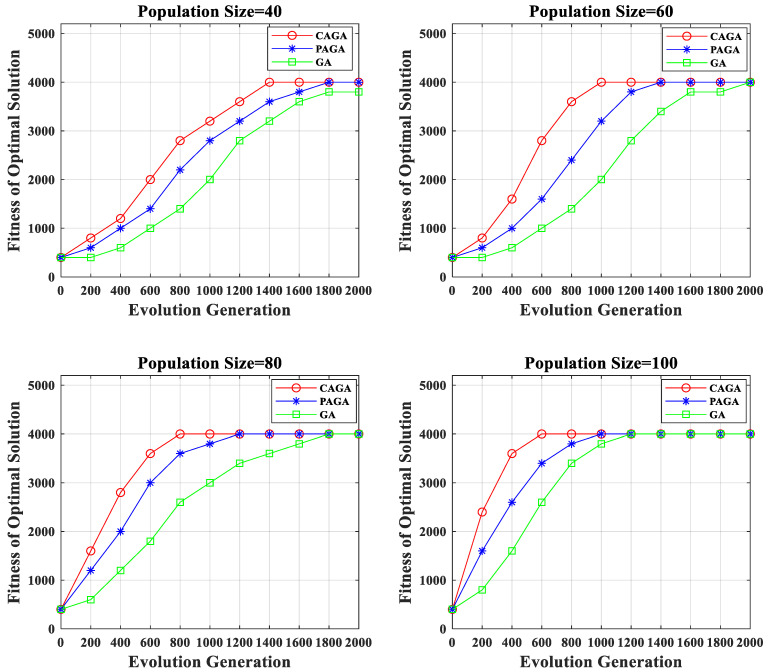
The performance of improved GA.

**Figure 9 sensors-22-07930-f009:**
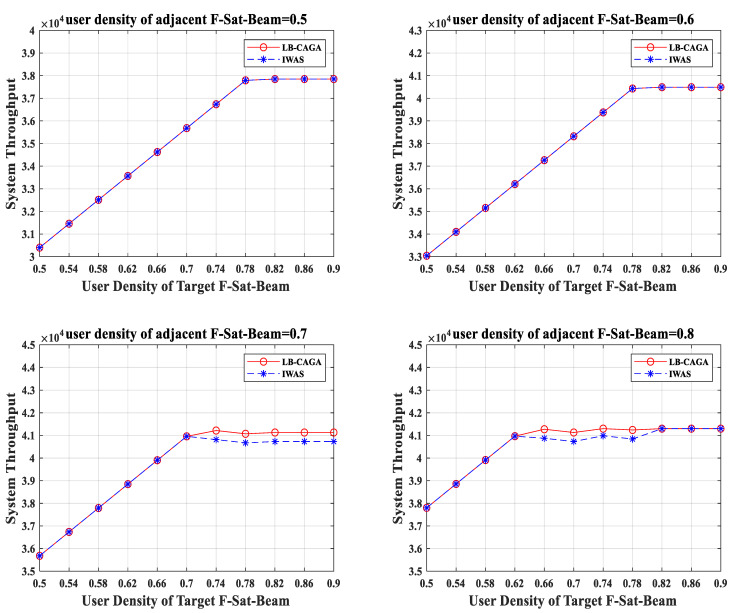
The performance of improved load balancing schemes.

**Table 1 sensors-22-07930-t001:** The solution modeling of load balancing by GA.

Genetic Algorithm Mode	Modeling of Satellite User Load Balancing Based on Genetic Algorithms
Fitness Function	The throughput of adjacent Sat-Beam $B_{i},B_{j}$ in the satellite service period $T_{i}$ and average waiting time for users of adjacent Sat-Beam $B_{i},B_{j}$.
Fitness Rules	The fitness increases with the increase of throughput and decreases with the increase of the average waiting time for users.
Single Optimal Solution	The load balancing scheme $A_{ij}$ of the local k Ser-Beams $b_{ij}$ when the user density is $p\left(b_{ij}\right)$ and the serving satellite is $S_{ij}$.
Solution Encoding	Natural number encoding.
Solution Code	$A_{ij}=\left\{a_{1}\;,a_{2}\;,a_{3}\;,\cdots a_{k}\right\}$, $a_{m}\;\left(1\leq m\leq k\right)$ denotes the access scheme of the m-th F-Sat-Beams.
Factors Influencing the Solution	The densities variances $∆p_{g}\left(b_{ij}\;,T_{i}\right)$ of DoU in the direction of SSPs moving in a Ser-Beam $b_{ij}$ and serving time $T_{i}$, differences with that of adjacent Ser-Beams $∆p_{g}\left(b_{ij}\;,b_{i+∆i\;j+∆j}\;,T_{i}\right)$.
Selected Set of Solutions	n load balancing schemes $A_{ij}^{1}\;,A_{ij}^{2}\;,A_{ij}^{3}\;,\cdots ,A_{ij}^{n}$ of local k Ser-Beams $b_{ij}$ when the user density is $p\left(b_{ij}\right)$ and the serving satellite is $S_{ij}$
A Set of Solutions Selected According to Fitness	The N schemes with the highest fitness among the existing load balancing schemes $A_{ij}^{1}\;,A_{ij}^{2}\;,A_{ij}^{3}\;,\cdots \;,A_{ij}^{N}$
The Process of Coding Crossover	Select two load balancing schemes $A_{ij}^{x}\;,A_{ij}^{y}$ with similar fitness according to the crossover probability. Randomly select the points for crossover and exchange the elements of the corresponding points of the two solutions.
The Process of Coding Mutation	Select a load balancing scheme $A_{ij}^{x}$ based on mutation probability. The mutation points are randomly selected, and the elements of the corresponding points of $A_{ij}^{x}$ are changed according to the selection probability in the value set.

**Table 2 sensors-22-07930-t002:** Simulation parameter configuration for GA.

	CAGA	PAGA	GA
**Crossover**	Two-point crossover	Two-point crossover	Two-point crossover
**Mutation**	Single-point mutation	Single-point mutation	Single-point mutation
**Select**	Tournament	Tournament	Tournament
**Crossover Probability**	Base probability 0.6	Base probability 0.6	fixed probability 0.6
**Mutation Probability**	Base probability 0.1	Base probability 0.1	fixed probability 0.1
**Population Size**	40/60/ 80/100	40/60/ 80/100	40/60/ 80/100
**Evolution Generation**	0:200:2000	0:200:2000	0:200:2000
**Elitist Preservation**	use	use	use
**Termination Condition**	Reach the maximum evolutionary generation	Reach the maximum evolutionary generation	Reach the maximum evolutionary generation

**Table 3 sensors-22-07930-t003:** Simulation parameters configuration of load balancing.

Number of F-Ser-Beam	9
Traffic Value of F-Ser-Beam	400M
User Density of Target F-Sat-Beam	0.5: 0.04: 0.9
User Density of Adjacent F-Sat-Beam	0.5/0.6/0.7/0.8
Satellite Total Bandwidth	2G
Average Traffic Value of User Request	15M
Average Arrival Rate of User Request	0.6

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
