# Peer review of "A Novel Load Balancing Scheme for Satellite IoT Networks Based on Spatial–Temporal Distribution of Users and Advanced Genetic Algorithms"

_sensors, 2022, doi:10.3390/s22207930_

Round 1
Reviewer 1 Report
In this article, the authors propose a novel load balancing scheme of adjacent beams for Satellite IoT Networks based on the modeling of spatial-temporal distribution of users and advanced genetic algorithm. The article has a contribution, however, the presentation is not good. The English written must be improved, the language is very informal. The article is a formal document. The use of " 's " or " ' " is not recommended. The article needs a complete review of English. The author should improve the presentation of the flow chart in figure 6. The font size must be the same as the main text. The authors should justify the choice of results parameters with articles that use a similar scenario. There are figures without commentaries of the results, all figures must be analyzed. Moreover, essential parts of the text are not well exploited; for instance, the authors cite that the proposed scheme calculates the PDF; however, the expression and details for the PDFs are not included. It is hard to follow the analysis in parts of the text.
Reviewer 2 Report
In the Summary, there is no information about the brain's ability to apply the research results in practice. According to the reviewer, there are no sources of their origin in the description of the drawings. Is it my own work, or is it based on bibliography, etc.
Reviewer 3 Report
The manuscript proposes a novel load balancing scheme of adjacent beams for S-IOT-N based on the modeling of 25 spatial-temporal users’ distribution and advanced GA.
The following comments to be addressed:
- The title is too long. It is recommended to make it short.
- The abstract includes some sentences which do not convey any useful information. For example the first sentence "Sensing and monitoring anywhere, anytime, anyway in more dimensions are quite 18 important for the coming era of Digital twins, metaverse." Why not introducing the problem and saying how do you address this problem?
- The last sentence of the abstract says "the simulations show that the proposed method r can effectively improve the average throughput ...." but how much improvement? talk with numbers, % , ....
- The related work section needs to be comprehensive. The following related work are recommended:
S. Abulgasem, F. Tubbal, R. Raad, P. I. Theoharis, S. Lu and S. Iranmanesh, "Antenna Designs for CubeSats: A Review," in IEEE Access, vol. 9, pp. 45289-45324, 2021, doi: 10.1109/ACCESS.2021.3066632.
C. Dong, X. Xu, A. Liu and X. Liang, "Load balancing routing algorithm based on extended link states in LEO constellation network," in China Communications, vol. 19, no. 2, pp. 247-260, Feb. 2022, doi: 10.23919/JCC.2022.02.020.
Liu, S.; Theoharis, P.I.; Raad, R.; Tubbal, F.; Theoharis, A.; Iranmanesh, S.; Abulgasem, S.; Khan, M.U.A.; Matekovits, L. A Survey on CubeSat Missions and Their Antenna Designs. Electronics 2022, 11, 2021. https://doi.org/10.3390/electronics11132021
H. Cao, Y. Su, Y. Zhou and J. Hu, "QoS Guaranteed Load Balancing in Broadband Multi-Beam Satellite Networks," ICC 2019 - 2019 IEEE International Conference on Communications (ICC), 2019, pp. 1-6, doi: 10.1109/ICC.2019.8761184.
Madni, M.A.A.; Iranmanesh, S.; Raad, R. DTN and Non-DTN Routing Protocols for Inter-CubeSat Communications: A comprehensive survey. Electronics 2020, 9, 482. https://doi.org/10.3390/electronics9030482
C. Dong, X. Xu, A. Liu and X. Liang, "Load balancing routing algorithm based on extended link states in LEO constellation network," in China Communications, vol. 19, no. 2, pp. 247-260, Feb. 2022, doi: 10.23919/JCC.2022.02.020.
W. Liu, Y. Tao and L. Liu, "Load-Balancing Routing Algorithm Based on Segment Routing for Traffic Return in LEO Satellite Networks," in IEEE Access, vol. 7, pp. 112044-112053, 2019, doi: 10.1109/ACCESS.2019.2934932.
S. M. Shahid, Y. T. Seyoum, S. H. Won and S. Kwon, "Load Balancing for 5G Integrated Satellite-Terrestrial Networks," in IEEE Access, vol. 8, pp. 132144-132156, 2020, doi: 10.1109/ACCESS.2020.3010059.
- For the modelling section, you may use a clear example to show the steps
- Is the simulation setting based on a benchmark?
Round 2
Reviewer 1 Report
The authors satisfactorily answered all the comments I made in the previous review.
Reviewer 3 Report
Thanks for addressing the comments.
No more comments